# Effect of orally administered cannabidiol oil on daily tonometric curve in healthy Italian Saddle horses

Marilena Bazzano, Fulvio Laus, Matteo Cerquetella, Andrea Spaterna, Andrea Marchegiani *

School of Biosciences and Veterinary Medicine, University of Camerino, Matelica, Italy

* andrea.marchegiani@unicam.it

## Abstract

### Background

Phytocannabinoids have the potential to lower intraocular pressure in both normal and glaucomatous eyes and they have been tested in different animal species, but not in the horse. The present paper describes the tonometric curve of healthy adult Italian Saddle horses after oral administration of cannabidiol oil (CBD).

### Methods

CBD 20% was administered orally (oily solution) at the dose of 1 mg/kg to 8 adult horses and intraocular pressure (IOP) was evaluated by tonometric curve. Data were then compared to those of the same horses obtained the day before (blank) CBD administration.

### Results

15 minutes after CBD administration, IOP (time zero 27.3 ± 2.1 mmHg right eye; 24.6 ± 2.3 mmHg left eye) started to decrease (19.5 ± 5.2 mmHg right eye; 20.8 ± 2.4 mmHg left eye) and 1 hour later CBD it reached the minimum level in all horses (11.4 ± 7.5 mmHg right eye; 9.5 ± 5.8 mmHg left eye), remaining statistically significantly lower than normal values for the entire observation period (8 hours; 12.0 ± 7.9 mmHg right eye; 11.9 ± 7.8 mmHg left eye).

### Conclusions

CBD 20% was effective to significantly reduce IOP in healthy adult Italian Saddle horses and may be an effective hypotensive agent to be implemented in case of primary or secondary glaucoma.

**Data availability statement:** Data are available as Supporting information.

**Funding:** The author(s) received no specific funding for this work.

**Competing interests:** The authors have declared that no competing interests exist.

## Introduction

Intraocular pressure (IOP) is the result of the balance between production and drainage of the aqueous humor [1]. Several physiological and pathological conditions have been associated in horses with IOP fluctuation and, as happened in other mammalian species, a daily variation of IOP has been ascertained also in equids [2]. Physiologically, IOP shows a circadian rhythm, being low during the dark phase and high during the light phase of the day, with a peak at the end of the light phase [3]. In addition, exercise has been shown to positively impact IOP of horses, which tends to be lower after exercise in Italian Saddle horses [4]. While the reduction in IOP is usually infrequent and not harmful for the condition of the eye, the increase of intraocular pressure above levels considered normal (glaucoma) is common and represents a major cause retinal damage in both animals and humans [5]. Although equine primary glaucoma is seldom encountered in clinical practice, secondary glaucoma can be virtually linked to almost all cases of ocular disease in horses, and equine recurrent uveitis is recognized as the main cause [6–8]. Glaucoma has attracted the interest of many research groups in both human and veterinary patients and attempts have been made to apply phytotherapy/ nutraceuticals as multimodal therapeutical approach to manage vision-threatening diseases [9]. As happens with the discovery of new drugs for the management of chronic and degenerative disease, the unveiling of the endocannabinoid system has represented a breakthrough for the understanding and management of different diseases, especially those inflammatory in nature [10–12]. This system, composed by endocannabinoids, cannabinoid receptors, and metabolizing enzymes, is a very complex cell signaling pathway found in all chordates, including horses, and able to modulate several physiological and pathological processes, found to be present also in ocular structures [13,14]. In humans, the activation of cannabinoid receptors 1(CB1) and 2 (CB2) have been proposed to play a role in the IOP reduction, although further insight are needed to better ascertain such effects [15]. In horses, CB1 and CB2 have been found in many neural tissues as hippocampus, basal ganglia, cortex, cerebellum, ileum and others, speculating a possible presence of such receptors also in the retina and neural tissue of uvea [16]. The ocular effects of *Cannabis sativa* and cannabinoids have been studied extensively in animals and humans over the last few decades and, despite a debated hypothetical ocular hypotensive effects, have generated significant interest [17–19]. Phytocannabinoids represent the main source of exogenous cannabinoid administered to both humans and animals to study their interaction with the hosts in specific form of disease. In last few years pharmacokinetics, efficacy and tolerability of many phytocannabinoid formulations, mainly containing cannabidiol have been also investigated in pets medication, not only for dermatological application [20] but also for the control of acute pain caused by ovariohysterectomy and chronic pain caused by osteoarthritis [21,22]. In horses, cannabinoids have been tested for the management of mechanical allodynia, second intention wound healing, chronic degenerative pain, and treatment for stereotypic behavior such as crib-biting [23–29] and subsequent studies have analyzed pharmacokinetic properties of cannabinoids in horses [30–35].

In human medicine, cannabinoids have been known to lower IOP since many years and have been shown to have beneficial effects in glaucoma patients beyond their IOP-lowering properties [36]. To date, no similar investigations have been carried out in horses and the aim of the present study was to evaluate the possible lowering effect on intraocular pressure of orally administered cannabidiol oil on healthy horses by the evaluation of daily tonometric curve before and after cannabidiol oil administration.

## Materials and methods

All the procedures conducted on animals were approved by Institutional Animal Welfare Committee (protocol number E81AC.19) of the University of Camerino and by the Ministry of Health (authorization number: 1021/2023-PR), in accordance with the Directive 2010/63/EU of the European Parliament on the protection of animals used for scientific purpose. The study was conducted in July 2024 in Central Italy. Supposing a possible difference in IOP of about 5 mmHg, (alpha = 0.05, beta 0.05, power = 0.95), the number of participants on which conduct observations is eight. Eight Italian Saddle horses (equally distributed between males and females), aged between 10 and 12 years and weigh between 380 and 420 kg, were enrolled in the study. All were considered healthy based on routine physical examination findings and the results of blood testing (hematology and mineral profile, energy profile, hepatic profile; exams executed within a week prior enrolment), which were within reference ranges reported for the species. The presence of eye disease as well as alterations in physical examination represented exclusion criteria. Horses underwent complete ophthalmological examination including fluorescein staining, Schirmer tear testing, slit lamp examination, direct and indirect ophthalmoscopy, and tonometry evaluation (Tonopen Vet, Reichert Inc., Depew, NY USA). The horses were stabled in individual boxes at the Veterinary Teaching Hospital of the University of Camerino (Italy) and fed polyphite hay at a rate of 2.5% of their body weight divided into three daily meals (at 07:00, 13:00 and 19:00), water was available ad libitum. Since a significant transient increase in IOP was observed when horses usually stabled in paddock are moved to medical barn, mainly due to the stress of the new environment, an acclimatization period of five days was considered to rule out any possible interference of the housing system and stress on intraocular pressure [37]. IOP evaluation was performed on the right and left eyes of each animal. Each horse was subjected to nine measurements of IOP (8 am, before cannabidiol oil administration and then after 15 minutes, 30 minutes, 60 minutes, 2 hours, 3 hours, 4 hours, 6 hours, and 8 hours) with Tonopen Vet (Reichert Inc., Depew, NY USA) and evaluated three times (with a wash-out period of eight days between each CBD administration, to avoid any possible residual effect of preparation on the second and third tonometric curve, as per [12]), using the same scheme. The order of horses and eyes examined were chosen in a randomized manner. All tonometric measurements were made by the same examiner and were taken within the stall of each horse. During tonometry, the head of each horse was kept in a neutral position without pressure on the jugular veins and the containment was carried out by the same collaborating investigator. Cannabidiol oil (galenical formulation containing 20% of CBD in medium-chain triglyceride) was administered orally at a dose of 1 mg/kg by collaborating investigators using a syringe, at 8am after first IOP measurement. To assess a possible effect of cannabidiol oil in lowering IOP, a daily tonometric curve was built for all horses enrolled in the study the day prior cannabidiol oil administration.

The data obtained were analyzed using a mixed-effect model in Graph Pad Prism 10 for Mac (GraphPad Software, Boston, Massachusetts USA, www.graphpad.com). Values of $p < 0.05$ were considered significant. This model allowed to analyze the effect of CBD administration, considering time and eye (right and left; fixed factors) and treating subjects as a random factor (to refer the data obtained to the general population and not only the population on which the study was carried out). In addition, a post-hoc analysis by Tukey's multiple comparisons test was carried out to highlight differences in IOP peak over time.

## Results

CBD oil was well tolerated by all patients that did not need twitch or similar restraint methods to perform tonometry.

Statistical analysis revealed no significant difference between IOP measurements at the three different times (p > 0.05); therefore, the mean values of the right and left eyes were considered in the following statistical analyses.

Mixed-effect model applied to the data showed a significant effect of CBD oil administration, time, and combined effect of CBD and time on IOP values (p < 0.0001 for all parameters). The mean IOP values for right and left eyes of horses at rest (prior CBD oil administration) and after CBD oil administration, expressed as mean ± standard deviations are reported in Table 1 and Fig 1. Table 1 also includes the statistically significant differences with respect to CBD, eye, and time points resulted from post-hoc analysis.

After CBD administration, IOP significantly started to decrease after 15 minutes (Fig 1) and then reached the minimum level one hour post CBD administration (especially in left eyes), maintaining this reduction consistently for the entire length of observations.

**Table 1. Mean values (± standard deviations) for IOP of right and left eyes of horses, before and after cannabidiol oil administration.**

|  | Before CBD oil administration | | After CBD oil administration | |
|---|---|---|---|---|
|  | **Right eye** | **Left eye** | **Right eye** | **Left eye** |
| Time zero | 27.3 ± 2.1 | 24.6 ± 2.3 | 27.3 ± 2.1 | 26.6 ± 2.3 |
| 15 min | 26.0 ± 1.7 | 26.0 ± 2.0 | 19.5 ± 5.2 | 20.8 ± 2.4[ABE] |
| 30 min | 26.0 ± 2.0 | 25.5 ± 2.2[FH] | 17.1 ± 4.0 CD | 16.6 ± 3.8[ABCDGEI] |
| 1h | 25.0 ± 1.5[J] | 25.0 ± 1.9[CDM] | 11.4 ± 7.5 CD | 9.5 ± 5.8[ABCDEFGIKLN] |
| 2h | 24.9 ± 1.5[BJM] | 24.4 ± 1.6[CDGM] | 11.1 ± 8.6[AB] | 12.8 ± 6.2[ABCDEGIL] |
| 3h | 24.6 ± 1.7[AJM] | 24.1 ± 1.6[CDM] | 11.4 ± 7.5[ACD] | 13.6 ± 5.9[ACDEGI] |
| 4h | 24.4 ± 1.6[ABEIJM] | 24.3 ± 1.7[CDM] | 11.5 ± 6.8[ABCDGKN] | 12.9 ± 6.3[ABCDEGIKN] |
| 6h | 23.6 ± 1.4[ABIM] | 23.6 ± 1.5[CDEGM] | 11.6 ± 6.7[ABCDEGIKN] | 12.3 ± 7.3[ABCD] |
| 8h | 23.5 ± 1.6[IM] | 23.6 ± 1.6[ACDGM] | 12.0 ± 7.9[B] | 11.9 ± 7.8 |

Values are expressed in mmHg. Superscripts capital letters indicate statistically differences between CBD and time points (A vs right eye after CBD time zero; B right eye before CBD time zero; C vs left eye after CBD time zero; D left eye before CBD time zero; E right eye before time zero 15 minutes; F left eye after CBD 15 minutes; G left eye before CBD 15 minutes; H right eye after CBD 30 minutes; I right eye before CBD 30 minutes; J left eye after CBD 3o minutes; K left eye before CBD 30 minutes; L right eye before CBD 1 hours; M left eye after CBD 1 hours; N left eye before CBD 1 hour).

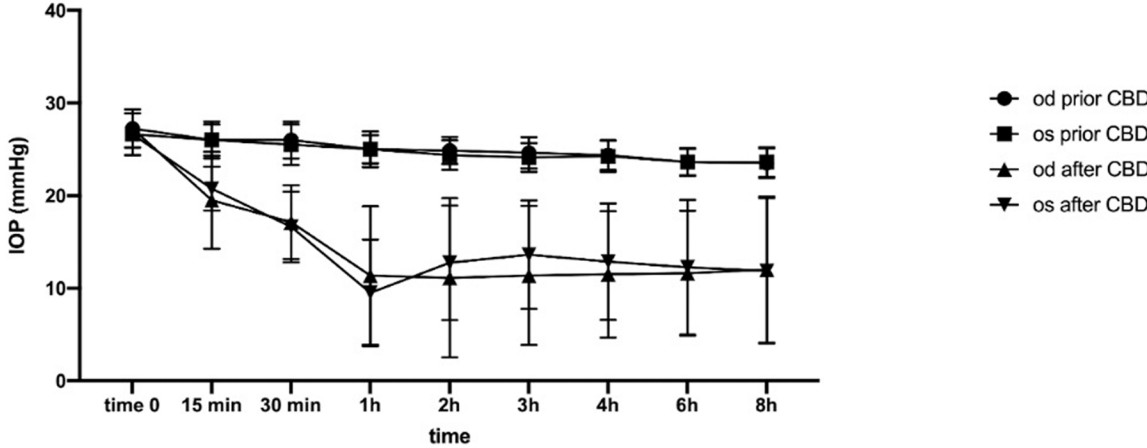

**Fig 1. Variation of IOP in right and left eyes at the different time points considering resting condition and after cannabidiol oil administration.** Since statistical analysis revealed no significant difference between IOP measurements at the three different times (p > 0.05), data (in mmHg) are expressed as mean values ± standard deviations of the three different evaluation times. Od: right eye; Os: left eye.

## Discussion

Intraocular pressure may vary (increase or decrease) depending on several physiological (i.e., exercise, fasting, circadian rhythm, etc.) and pathological (i.e., uveitis, inflammation, etc.) conditions [4,37–39].

The comparison between the left and right eyes of the horses included in the present study showed no differences in IOP at resting condition, which was within the physiological range indicated the species [8] while CBD administration was remarkably responsible for a considerable reduction of IOP soon after its administration.

Cannabidiol is one of over more than one hundred phytocannabinoids isolated from the Cannabis sativa plant which nowadays represents the main appealing novel therapeutic approach to manage different diseases in both animals and humans, including glaucoma [40]. Starting from 1970s, when *Cannabis sativa* was started to be studied for its medical purposes (and not considering its psychotropic effects) many different compounds have been uncovered and in early 1990s the insight into mechanism of action of phytocannabinoids has led to the identification of the main receptors [14,41]. In fact, the main therapeutic potential of endocannabinoid system passes through the binding between cannabinoids and their metabolites with cannabinoid receptors, CB1 and CB2, but also with non-cannabinoid receptors [36]. From a pharmacological point of view, CB1 and CB2 are receptors belonging to the G protein-coupled family; they can be positively stimulated by mitogen-activated protein kinase (MAPK) and negatively stimulated via adenylyl cyclase [42]. Few pharmacokinetic studies conducted in dogs, cats, and horses have evaluated that CBD and endocannabinoid system are able to exert anti-inflammatory, relaxing, anticonvulsant and anxiolytic effects [29]. Such results have been confirmed by clinical studies that investigated CBD administration for the treatment of osteoarthritis, canine epilepsy and canine atopic dermatitis with positive outcomes [43,44]. Despite deep insights into veterinary species are still lacking regarding the presence and distribution of CB receptors, in the human eye, CB1 receptors have been detected in several ocular structure such as the cornea, iris, ciliary body (including the epithelium, ciliary muscle and blood vessels of the ciliary body), trabecular meshwork, Schlemm's canal, and retina [45]. The location and precise functions of CB2 receptors in the human eye appear to be less detailed; they have been found in the cornea, trabecular meshwork, and retina, particularly in animal models [36]. This anatomical dissemination of CB1 receptors indicates that cannabinoids may influence IOP by both increasing aqueous humor outflow and decreasing aqueous humor production [45]. The mechanism of action of cannabinoids has not been fully elucidated yet and many theories have been proposed. The presence of CB1 receptors in trabecular meshwork and in Schlemm canal may explain the possible influence of cannabinoids on conventional aqueous humor outflow [46]. Aqueous humor is physiologically produced by uveal structure (ciliary body, epithelium, ciliary muscle) and normally drained by trabecular meshwork and, in minor portion, by uveoscleral outflow; the presence of CB1 receptors in this district may increase uveoscleral outflow and thus decrease IOP, and represents a clear starting point for the study and rational use of CDB for IOP management [47]. Interestingly, cannabinoid receptors have been also located in retinal structures [48]. In fact, CB1 receptors have been isolated in the ganglion layer, the inner and outer plexiform layers, the inner nuclear layer, and the outer segments of photoreceptors [48]. CB2 receptors have also been described in retinal cell types and layers, like amacrine, bipolar, Müller, microglial, and RGCs, or in the retinal pigment epithelium [48]. The presence of CBD receptors in different retinal cells and district may represent the start point for the study and application of CBD to manage different inflammatory and degenerative retinal conditions.

A body of work has been conducted in veterinary patients to assess the potential of the use of CBD for ophthalmic diseases, mainly for its anti-inflammatory and neuroprotective effects (i.e., IOP regulation during glaucoma, corneal and uveal diseases and retinal/optic nerve head diseases) [44]. IOP lowering effects of phytocannabinoids in different animal species, including rabbits, mice, cats, dogs, and monkeys have been studied with controversial results [42]. Some authors indicate that CBD has no effect on IOP other authors found a decrease in IOP after CBD administration, irrespective of the route of administration; other authors found a CBD-induced increase in IOP [42].

The results of the present study are the first, at the best of authors' knowledge, that aimed to appraise an effect of CBD on IOP in healthy equine subjects. The major effect of CBD on left eyes may be due to a normal difference in IOP measurement between right and left eyes, ad ascertained in humans but not yet in horses [49].

One of the main limitations of the present study is the lack of a clearly depicted presence of CB receptors in equine eye structure that limits only to speculation and assumption the mechanism of action of CBD oil in production and drainage of aqueous humor. This aspect needs to be deepened in detail to clearly explain the physiological basis of such an effect, possibly open new frontiers in therapeutic potential of CBD in equine eye. All the evaluations have been done in a specific horse breed, and it would be interesting to extend such investigation in other breeds that may respond differently to CBD.

## Conclusion

Intraocular pressure is the most important modifiable risk factor for glaucoma, that is a labile equilibrium between production and drainage, that can be easily disrupted by any ephemeral stimulus causing a fluctuation that may be risky for the health of the eye. Glaucoma is worldwide recognized to be a neuropathy caused by an increase in IOP that affects the retina, and the presence, hypothesized at this time but needed to be confirmed, of retinal cannabinoid receptors may explain the functional modulation of neuroretinal cells in animal tissues. The results of the present study are only a little step forward the fascinating world of endocannabinoid system modulation for IOP management, therefore posing a potential target for neuroprotection and glaucoma control that deserve to be deepened in detail.

## Supporting information

**S1 File. Supporting informations.**
(XLSX)

## Author contributions

**Conceptualization:** Andrea Spaterna, Andrea Marchegiani.

**Data curation:** Marilena Bazzano, Fulvio Laus.

**Investigation:** Marilena Bazzano, Fulvio Laus, Andrea Marchegiani.

**Methodology:** Marilena Bazzano, Matteo Cerquetella, Andrea Marchegiani.

**Writing – original draft:** Marilena Bazzano, Fulvio Laus, Matteo Cerquetella.

**Writing – review & editing:** Andrea Spaterna, Andrea Marchegiani.

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
