## [Decision Letter · Decision Letter 0]

21 Feb 2025

PONE-D-25-04742Effect of orally administered cannabidiol oil on daily tonometric curve in healthy Italian Saddle horsesPLOS ONE

Dear Dr. Marchegiani,

Thank you for submitting your manuscript to PLOS ONE. After careful consideration, we feel that it has merit but does not fully meet PLOS ONE’s publication criteria as it currently stands. Therefore, we invite you to submit a revised version of the manuscript that addresses the points raised during the review process.

We look forward to receiving your revised manuscript.

Kind regards,

Claudia Interlandi, Ph.D

Academic Editor

PLOS ONE

Comments from PLOS Editorial Office:

We note that one or more reviewers has recommended that you cite specific previously published works. As always, we recommend that you please review and evaluate the requested works to determine whether they are relevant and should be cited. It is not a requirement to cite these works. We appreciate your attention to this request.

2. To comply with PLOS ONE submissions requirements, in your Methods section, please provide additional information regarding the experiments involving animals and ensure you have included details on (1) methods of anesthesia and/or analgesia, and (2) efforts to alleviate suffering.

Additional Editor Comments (if provided):

The reviewers agree that many parts of the manuscript are in need of deepening and major revision.

Reviewers' comments:

Reviewer's Responses to Questions

**Comments to the Author**

1. Is the manuscript technically sound, and do the data support the conclusions?

Reviewer #1: Partly

Reviewer #2: No

2. Has the statistical analysis been performed appropriately and rigorously? 

Reviewer #1: Yes

Reviewer #2: No

3. Have the authors made all data underlying the findings in their manuscript fully available?

Reviewer #1: Yes

Reviewer #2: Yes

4. Is the manuscript presented in an intelligible fashion and written in standard English?

Reviewer #1: Yes

Reviewer #2: Yes

5. Review Comments to the Author

Reviewer #1: Please see the attached file.

The manuscript entitled “Effect of orally administered cannabidiol oil on daily tonometric curve in healthy Italian Saddle horses.” investigate the use of a natural substance such as cannabis oil to evaluate the effects of such administration on eye pressure trends in the horse. This work is very interesting, in terms of evaluating the health status of the horse's eye but also as it relates to the diagnostic aspect of eye diseases. It is well structured but some parts turn out to be written in too superficial and shallow a manner to be a scientific work.

In my opinion this work should be deepened and revised in many parts, and rewritten in the discussion, also following the advice and comments below following a major revision.

Reviewer #2: I have read and reviewed the manuscript entitled: Effect of orally administered cannabidiol oil on daily tonometric curve in healthy Italian Saddle horses. Overall, from this reviewer’s perspective, this is an interesting study; however, in its current state, it shows many deficiencies. For example, the methodology section requires an orderly and detailed description to ensure that the study can be replicated. Regarding the statistical analysis, I suggest the authors modify it to a mixed linear model that allows the establishment of statistically significant differences between treatments and the various measurement moments, in addition to correctly analyzing the tonometric curve on the three occasions in which the measurements were repeated. On the other hand, the lack of a control group generates an important bias in the research. Finally, I would like to highlight that the discussion section should be rewritten. This is to explain the neurophysiological events that occur under the effect of CBD to reduce intraocular pressure.

Likewise, other observations must be addressed to achieve publication quality. I have left some comments, hoping that they can help the authors.

General comments

L31-33: Please indicate the IOP values before and after CBD administration, with their respective P values.

L66: CBD has also been used in pets for the control of acute pain caused by an ovariohysterectomy and chronic pain caused by osteoarthritis. Please add the following references to your manuscript, which you can also use in the discussion of your manuscript:

10.3389/fvets.2024.1380022

10.3389/fvets.2022.1050884

L69: I suggest that the authors add the following references in this line:

10.3389/fvets.2024.1496473

10.3389/fvets.2024.1341396

10.1016/j.jevs.2019.102880

L73: Please add a hypothesis.

L82: What was the statistical method used to determine the sample size? Please clarify.

L84: Please indicate which exclusion criteria were considered in your study.

L85: What were the analytes or parameters measured in blood before the start of the study? And how long before the administration of CBD were these blood studies performed?

L108: Due to the type of experimental design used, I suggest the authors perform a statistical analysis based on a linear mixed model instead of a 2-way ANOVA. This model will allow the analysis of the differences observed between treatments and evaluation times, and even for the analysis of the tonometric curve (comparison of three measurements) with their respective washout periods.

L123: Table 1, please indicate the statistical differences observed between treatments and evaluation times with letters and numbers. In this sense, also the results obtained after the administration of CBD at 15 minutes, 30 minutes, 60 minutes, 2 hours, 3 hours, 4 hours, 6 hours, and 8 hours should be integrated into this table; although Figure 1 already shows this information.

L141: In general, the discussion is a section that is shown to be deficient. Therefore, I suggest that the authors complement this section. For example, the neurophysiological effects that generate the decrease in intraocular pressure under the effect of CBD should be explained. I suggest consulting and citing the following references:

10.2460/javma.24.06.0360

10.3934/Neuroscience.2024009

10.3390/ph17060748

10.1016/j.ijpharm.2022.121627

10.3390/ijms22073798

10.1016/j.survophthal.2020.07.002

10.1097/01.ijg.0000212260.04488.60

10.3389/fvets.2022.1050884

10.3390/ph16081149

10.1016/j.biopha.2022.112981

10.1055/a-1665-3100

10.1016/j.exer.2020.108266

10.2174/1570159X15666170724104305

10.1016/j.biopha.2016.11.106

10.1155/2016/9364091

10.1016/S0079-6123(08)01131-X

https://pubmed.ncbi.nlm.nih.gov/34283478/

L167: I suggest that the authors discuss the limitations and perspectives of their study.

L171: Conclusions must be rewritten based on changes arising from modifications made to the statistical analysis.

6. PLOS authors have the option to publish the peer review history of their article (what does this mean? ). If published, this will include your full peer review and any attached files.

**Do you want your identity to be public for this peer review?** For information about this choice, including consent withdrawal, please see our Privacy Policy .

Reviewer #1: No

Reviewer #2: No

---

## [Author Response · Author response to Decision Letter 0]

20 Mar 2025

Below, in bold blue type, the point-to-point reply to reviewers

Reviewer #1: Please see the attached file.

The manuscript entitled “Effect of orally administered cannabidiol oil on daily tonometric curve in healthy Italian Saddle horses.” investigate the use of a natural substance such as cannabis oil to evaluate the effects of such administration on eye pressure trends in the horse. This work is very interesting, in terms of evaluating the health status of the horse's eye but also as it relates to the diagnostic aspect of eye diseases. It is well structured but some parts turn out to be written in too superficial and shallow a manner to be a scientific work.

In my opinion this work should be deepened and revised in many parts, and rewritten in the discussion, also following the advice and comments below following a major revision.

The manuscript entitled “Effect of orally administered cannabidiol oil on daily tonometric curve in healthy Italian Saddle horses.” investigate the use of a natural substance such as cannabis oil to evaluate the effects of such administration on eye pressure trends in the horse. This work is very interesting, in terms of evaluating the health status of the horse's eye but also as it relates to the diagnostic aspect of eye diseases. It is well structured but some parts turn out to be written in too superficial and shallow a manner to be a scientific work.

In my opinion this work should be deepened and revised in many parts, and rewritten in the discussion, also following the advice and comments below following a major revision.

Introduction

Line 49 Please unify the number of bibliographical sources and dots through the main text Text has been modified to reflect this comment

Line 45-54 As the title of the present article suggest that you were talking about healthy subjects, first of all I suggest to describe the physiological oscillation of the IOP for example after exercise in horses, for example include this source to deep the section:

doi: 10.3390/ani12141850. Text has been modified to reflect this comment and reference has been added

Line 49-55 this section is described too broadly and in too little depth, so the arguments regarding therapeutic approaches are disconnected with the introductory section Section rephrased to ensure more clarity

Line 56-57 some new bibliographic source to accompany the section should be included as:

https://doi.org/10.3389/fvets.2024.1496473

https://doi.org/10.3389/fvets.2024.1341396

Text has been modified to reflect this comment and references have been added

Materials and methods

Line 91-92 if the horses were already stabled in the mentioned way, according to precise routines, what was the purpose of the 5-day acclimatization period? Please be more specific and specify what the horses needed to acclimatize to Lines implemented in the manuscript and a proper reference has been added

Line 92-93 With what instrumentation the IOP was measured and by what methodology IOP was evaluated with Tonopen Vet, Reichert Inc., Depew, NY USA. Detailed in the text

Line 101-102 The product manufacturer should be reported

Results and Discussion

Line 116-117 data were normalized results or not? this result should be reported

Line 135-136 The mention both eyes and respectively right and left is redundant

Line 137 the same mention “No effect of sex was found” is redundant compared to lines 133-134

Line 116-121 In addition, a two-way ANOVA was performed in the statistical analysis. Better clarify which variables were evaluated. Also, in the results section these significant values should be detailed. To establish a significant peak over time, a post hoc analysis should also have been performed, so describe it accurately both in the statistical analysis section and in the results.

Results in too little content. According to when IOP measurements over time were assessed before CBD administration and then 3 times every 8 days. Why does it appear in Figure 1 only before and after administration. Please clarify these concepts so that they are easy for a reader to interpret.

Table 1 If results were calculated and expressed as mean and standard deviation, this information should be specified in the results rection

Line 143-144 Avoid repetitions

Line 151 Please mention the examples reported in horses as showed in the previously suggested bibliographical sources

Line 155 nothing was previously mentioned about THC use or information in the text. Please explain its importance and usage even in other sections or delete it.

The importance of eye pressure changes in both physiological and pathological settings should be emphasized.

The discussions are too short and poor in content. there is described only what has been found in the literature on the topic. The discussion section should be a discussion regarding the results obtained, so one should first describe what was obtained, comment on its possible meanings, and discuss comparison with other studies that found similar or opposite results based on the variables analyzed, so the section is all in need of rewriting

Conclusions should also more emphasize the results obtained and indicate in what ways these results can be useful in the scientific community

Statistical analysis has been run again, applying both a mixed effect model analysis and post hoc analysis to reflect your recommendations, so the paragraph has been rewritten accordingly. Results, Discussion and Conclusion sections have been completely rewritten to be more specific and consistent, incorporating all your suggestion

Reviewer #2: I have read and reviewed the manuscript entitled: Effect of orally administered cannabidiol oil on daily tonometric curve in healthy Italian Saddle horses. Overall, from this reviewer’s perspective, this is an interesting study; however, in its current state, it shows many deficiencies. For example, the methodology section requires an orderly and detailed description to ensure that the study can be replicated. Regarding the statistical analysis, I suggest the authors modify it to a mixed linear model that allows the establishment of statistically significant differences between treatments and the various measurement moments, in addition to correctly analyzing the tonometric curve on the three occasions in which the measurements were repeated. On the other hand, the lack of a control group generates an important bias in the research. Finally, I would like to highlight that the discussion section should be rewritten. This is to explain the neurophysiological events that occur under the effect of CBD to reduce intraocular pressure.Likewise, other observations must be addressed to achieve publication quality. I have left some comments, hoping that they can help the authors.

Thanks for your insights and comments. We have amended the text in the proper parts to reflect your suggestions.

General comments

L31-33: Please indicate the IOP values before and after CBD administration, with their respective P values. Text modified to reflect this comment

L66: CBD has also been used in pets for the control of acute pain caused by an ovariohysterectomy and chronic pain caused by osteoarthritis. Please add the following references to your manuscript, which you can also use in the discussion of your manuscript:

10.3389/fvets.2024.1380022

10.3389/fvets.2022.1050884

Text modified to reflect this comment

L69: I suggest that the authors add the following references in this line:

10.3389/fvets.2024.1496473

10.3389/fvets.2024.1341396

10.1016/j.jevs.2019.102880

Text modified to reflect this comment

L73: Please add a hypothesis. Text modified to reflect this comment

L82: What was the statistical method used to determine the sample size? Please clarify.

The sample size needed was calculated with Sealed Envelope™ calculator, setting significance level at 5% and power at 95%, detailed in the text

L84: Please indicate which exclusion criteria were considered in your study. Text modified to reflect this comment

L85: What were the analytes or parameters measured in blood before the start of the study? And how long before the administration of CBD were these blood studies performed? Text modified to reflect this comment

L108: Due to the type of experimental design used, I suggest the authors perform a statistical analysis based on a linear mixed model instead of a 2-way ANOVA. This model will allow the analysis of the differences observed between treatments and evaluation times, and even for the analysis of the tonometric curve (comparison of three measurements) with their respective washout periods. Statistical analysis has been run according to your suggestion and text modified to reflect this comment

L123: Table 1, please indicate the statistical differences observed between treatments and evaluation times with letters and numbers. In this sense, also the results obtained after the administration of CBD at 15 minutes, 30 minutes, 60 minutes, 2 hours, 3 hours, 4 hours, 6 hours, and 8 hours should be integrated into this table; although Figure 1 already shows this information. Table 1 has been updated to reflect your comment

L141: In general, the discussion is a section that is shown to be deficient. Therefore, I suggest that the authors complement this section. For example, the neurophysiological effects that generate the decrease in intraocular pressure under the effect of CBD should be explained. I suggest consulting and citing the following references:

10.2460/javma.24.06.0360

10.3934/Neuroscience.2024009

10.3390/ph17060748

10.1016/j.ijpharm.2022.121627

10.3390/ijms22073798

10.1016/j.survophthal.2020.07.002

10.1097/01.ijg.0000212260.04488.60

10.3389/fvets.2022.1050884

10.3390/ph16081149

10.1016/j.biopha.2022.112981

10.1055/a-1665-3100

10.1016/j.exer.2020.108266

10.2174/1570159X15666170724104305

10.1016/j.biopha.2016.11.106

10.1155/2016/9364091

10.1016/S0079-6123(08)01131-X

https://pubmed.ncbi.nlm.nih.gov/34283478/

L167: I suggest that the authors discuss the limitations and perspectives of their study.

L171: Conclusions must be rewritten based on changes arising from modifications made to the statistical analysis.

Discussion and Conclusion sections have been completely rewritten to be more specific and consistent, incorporating all your suggestion

---

## [Decision Letter · Decision Letter 1]

22 Apr 2025

PONE-D-25-04742R1Effect of orally administered cannabidiol oil on daily tonometric curve in healthy Italian Saddle horsesPLOS ONE

Dear Dr. Marchegiani,

Thank you for submitting your manuscript to PLOS ONE. After careful consideration, we feel that it has merit but does not fully meet PLOS ONE’s publication criteria as it currently stands. Therefore, we invite you to submit a revised version of the manuscript that addresses the points raised during the review process.

We look forward to receiving your revised manuscript.

Kind regards,

Claudia Interlandi, Ph.D

Academic Editor

PLOS ONE

Journal Requirements:

Additional Editor Comments:

The manuscript needs further revision before it can be considered for publication.

Reviewers' comments:

Reviewer's Responses to Questions

**Comments to the Author**

1. If the authors have adequately addressed your comments raised in a previous round of review and you feel that this manuscript is now acceptable for publication, you may indicate that here to bypass the “Comments to the Author” section, enter your conflict of interest statement in the “Confidential to Editor” section, and submit your "Accept" recommendation.

Reviewer #3: (No Response)

2. Is the manuscript technically sound, and do the data support the conclusions?

Reviewer #3: Yes

3. Has the statistical analysis been performed appropriately and rigorously? 

Reviewer #3: I Don't Know

4. Have the authors made all data underlying the findings in their manuscript fully available?

Reviewer #3: Yes

5. Is the manuscript presented in an intelligible fashion and written in standard English?

Reviewer #3: No

6. Review Comments to the Author

Reviewer #3: I reviewed the edited version of the manuscript and provide the following comments to the authors. The majority of my comments are intended to correct the manuscript for grammatical errors and clarity.

Line 52/53 – suggest changing to “…common and represents a major cause of retinal….

Line 65 – suggest rewording the section “determine a lowering in IOP” to be more clear and understandable.

Line 74 – 76 – suggested edit “In the last few years…investigated in pet medication, …”

Line 116 – how was the length of the washout period determined? This information should be added to the manuscript.

Line 165 – suggested edit “….after one hour post CBD administration…left eyes) and maintaining….”

Line 178 – 180 – this sentence should be reworded to expand and provide clarity on the intended information this sentence is adding to the manuscript.

Line 188 – suggested edit “…was started to be studied for its…”

Line 190 – edit to “…the insight into the mechanism….”

Line 201 – 207 – this section needs to be reworded to ensure clarity for the reader.

Line 231 – 233 – please expand this section so that the point you are making is clear to the reader.

7. PLOS authors have the option to publish the peer review history of their article (what does this mean? ). If published, this will include your full peer review and any attached files.

**Do you want your identity to be public for this peer review?** For information about this choice, including consent withdrawal, please see our Privacy Policy .

Reviewer #3: No

---

## [Author Response · Author response to Decision Letter 1]

5 May 2025

Dear Editor and Reviewer, on behalf of all other Authors, I would thank for the time and efforts spent in reviewing the manuscript and for the points raised, which allow to improve the overall quality of the paper. We have carefully considered your suggestions and addressed them throughout thew text, using the revision function of Microsoft Word to highlight changes in the text.

In addition, English language has been reviewed by a native speaker.

Below, in bold blue type, the point-to-point reply to reviewers

I reviewed the edited version of the manuscript and provide the following comments to the authors. The majority of my comments are intended to correct the manuscript for grammatical errors and clarity.

Line 52/53 – suggest changing to “…common and represents a major cause of retinal…. Text has been modified to reflect this comment

Line 65 – suggest rewording the section “determine a lowering in IOP” to be more clear and understandable. Text has been modified to reflect this comment

Line 74 – 76 – suggested edit “In the last few years…investigated in pet medication, …” Text has been modified to reflect this comment

Line 116 – how was the length of the washout period determined? This information should be added to the manuscript. Detailed in the text

Line 165 – suggested edit “….after one hour post CBD administration…left eyes) and maintaining….” Text has been modified to reflect this comment

Line 178 – 180 – this sentence should be reworded to expand and provide clarity on the intended information this sentence is adding to the manuscript. Text has been modified to reflect this comment

Line 188 – suggested edit “…was started to be studied for its…” Text has been modified to reflect this comment

Line 190 – edit to “…the insight into the mechanism….” Text has been modified to reflect this comment

Line 201 – 207 – this section needs to be reworded to ensure clarity for the reader. Text has been modified to reflect this comment

Line 231 – 233 – please expand this section so that the point you are making is clear to the reader. Text has been modified to reflect this comment

---

## [Editor Report · Decision Letter 2]

9 May 2025

Effect of orally administered cannabidiol oil on daily tonometric curve in healthy Italian Saddle horses

PONE-D-25-04742R2

Dear Dr. Andrea Marchegiani,

We’re pleased to inform you that your manuscript has been judged scientifically suitable for publication and will be formally accepted for publication once it meets all outstanding technical requirements.

Kind regards,

Claudia Interlandi, Ph.D

Academic Editor

PLOS ONE

---

## [Editor Report · Acceptance letter]

PONE-D-25-04742R2

PLOS ONE

Dear Dr. Marchegiani,

I'm pleased to inform you that your manuscript has been deemed suitable for publication in PLOS ONE. Congratulations! Your manuscript is now being handed over to our production team.

Kind regards,

on behalf of

Professor Claudia Interlandi

Academic Editor

PLOS ONE